# A LABEL IS A LABEL IS A LABEL: RELATION AUGMENTATION FOR SCENE GRAPH GENERATION

## ABSTRACT

The goal of scene graph generation (SGG) is to identify the relationships between objects in an image. Many recent methods have been proposed to address a critical challenge in SGG - the biased distribution of relations in the dataset and the semantic space. Although the unbiased SGG problem has recently gained popularity, current SGG research has not thoroughly examined different types of augmentation. Recent works have focused on augmenting objects instead of relations and ignored the opportunities for pixel-level augmentation. We propose a novel relation augmentation method to use semantic and visual perturbations to balance the relation distribution. We use relation dataset statistics to boost the distribution of rare relation classes. We also use visual MixUp and grafting techniques to increase the sample size of triplets with tail relation labels. Our proposed method, RelAug, effectively reduces the long-tail distribution of predicates. We demonstrate this method can be easily adapted to existing methods and produces state-of-the-art performance on the Visual Genome dataset. The authors will make the source code publicly available for reproduction.

## 1 INTRODUCTION

"Rose is a rose is a rose is a rose." In her poem Sacred Emily, Gertrude Stein famously equates a label with its imagery and emotions (Stein, 1922). Equally rich with imagery and labels is the domain of scene graph generation (SGG). A *scene graph* like the one in Figure 3 succinctly captures the gist of an image by labeling the relations among objects. With such a powerful ability to represent visual knowledge, SGs have understandably seen wide adoption in vision-and-language tasks such as image captioning (Wu et al., 2017; Gu et al., 2018; Yang et al., 2019), visual question answering (VQA) (Fader et al., 2014; Gu et al., 2019; Wang et al., 2017; Wu et al., 2017; Zhao et al., 2018), and image retrieval (Johnson et al., 2015; Schuster et al., 2015). Despite its wide applications, one challenge limits the potential of SGG. As (Zellers et al., 2018) first points out, a long-tailed distribution of relation labels hinders SGG models from achieving acceptable performance. In fact, simply guessing the most frequent relation labels outperforms many sophisticated neural network models (Zellers et al., 2018; Tang et al., 2020). Figure 1 illustrates this long-tail bias of the relation labels in the Visual Genome dataset (Krishna et al., 2017). This thorny dataset bias further limits the very semantic power for which SGs are designed. (Guo et al., 2021) argue that without addressing the lack of diversity in semantic labels, SGs convey less information by preferring more common but less informative relations such as "on" when "standing on" would produce a richer visual and semantic context. The SGG research community has been seeking a remedy in several directions, including data rebalancing (Guo et al., 2021), logit adjustment (Guo et al., 2021), and incorporating external knowledge (Zareian et al., 2020). Others have attempted to apply triplet data augmentation to SGG. For SGG to generalize better on unseen triplets, Knyazev et al. generated new triplets by perturbing the objects and generating new scene graph "samples" in the latent space using Generative Adversarial Networks (Knyazev et al., 2021). Additionally, recent work by Zhang et al. has successfully applied MixUp to SGG triplet augmentation (Zhang et al., 2022). However, these approaches have missed an easy target for data augmentation, pixel-level relation augmentation.

Pixel-level visual data augmentation techniques, such as MixUp and CutMix (Zhang et al., 2018; Yun et al., 2019), have proven effective in computer vision tasks such as object detection. These methods create new visual examples by drawing from other sample images or image regions. Specifically, MixUp smooths two images, and CutMix grafts an image region onto another. In these methods, the

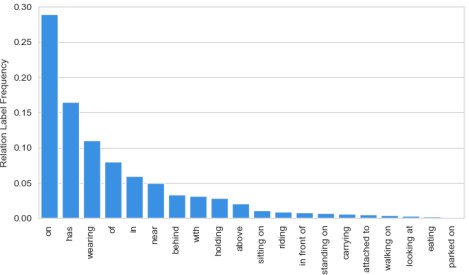 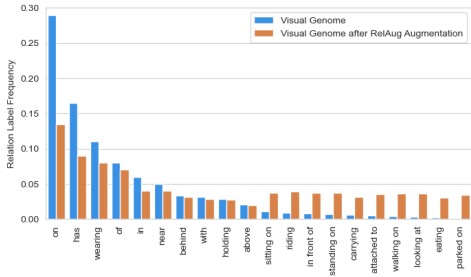

Figure 1: Our proposed vision-and-language augmentation technique, RelAug, address the long-tailed distribution of relation labels in the Visual Genome dataset (left) by increasing the label distribution of the least frequent labels (right).

generated samples share the same bounding boxes and object labels as the original. The empirical effectiveness of such pixel-level augmentation techniques serves as an inspiration for pixel-based relation perturbation. However, visual MixUp/CutMix cannot be easily applied to SGG. SGG requires abstract relational reasoning, that is, classifying the relationship between two objects or image regions. Admittedly, visual relations are secondary semantic representations that do not have a clear visual correspondence to the image. After all, what exactly does "standing on" look like? Conveniently, we do not have to solve this relation-to-pixel augmentation directly. Instead, we could proxy a relation-to-pixel augmentation by upsampling triplets containing such relations. Because a relation triplet contains both objects and relations, we can achieve relation augmentation by augmenting the objects in triplets. To close the sizeable recall-mean recall gap, we must focus on upsampling the tail relations to solve the long-tailed relation bias problem.

Our visual relation data augmentation approach takes advantage of both semantic and visual information. First, we use the semantic information to perturb the relation labels. Semantically, rather than perturbing the objects, we directly alter the relations. By retaining the same bounding boxes and object labels, we introduce new examples in which the relation label is substituted with another, sourced from various correlations in the training dataset. These correlations encompass random, word similarity, and cooccurrence statistics. This method has empirically demonstrated its effectiveness in enhancing categorical recall for the targeted tail relations. Visually, we aim to augment the sample size of triplets that feature tail relations. When presented with a scene graph and its subject or object, we seamlessly integrate another instance of that object into the designated image region. This strategy has been instrumental in amplifying the diversity of tail relations within the training dataset and rectifying biases in long-tailed relations. Drawing inspiration from our introductory poem, we pose a thought: if an object label remains consistent as an object label, shouldn't a relation label also maintain its essence as a relation label?

In summary, we propose the following contributions:

- We propose a novel augmentation strategy, relation augmentation (RelAug), to upsample second-order relation labels between visual objects in scene graphs.

- We develop a novel vision-and-language technique to generate new samples in the pixel domain using visual and semantic information.

- We demonstrate that our augmentation methods can be easily adapted to existing methods and produce state-of-the-art performance on the Visual Genome dataset. We examine the effect of augmentation on the SGG task and share the resulting augmented dataset.

We organize this work into the following sections. Section 2 contextualizes this work with regard to Unbiased Scene Graph Generation (SGG) and data augmentation. 3 will introduce our two methods for augmenting tail relations in scene graphs. 4 will introduce our primary dataset, Visual Genome (Krishna et al., 2017), and the three SGG task protocols: Predicate Classification (PredCls), Scene Graph Classification (SGCls), and Scene Graph Generation (SGGen). It will also introduce other experiment settings, such as hyperparameters and variants. Section 4.3 reviews experiment results

and ablation studies, providing quantitative and qualitative analysis of our methods. Finally, Section 5 summarizes the contributions and limitations of our work and assesses the impact of our research.

## 2 RELATED WORKS

### 2.1 UNBIASED SCENE GRAPH GENERATION

Earlier works in scene graph generation (SGG) such as IMP (Xu et al., 2017), VTransE (Zhang et al., 2017) and MotifNet (Zellers et al., 2018) have achieved high overall recall performance by innovating on the relation reasoning component of the neural network pipeline. However, recently works such as VCTree (Tang et al., 2019) and KERN (Chen et al., 2019) have pointed out an important caveat to such seemingly high performance brought about by deep neural architectures. Specifically, they highlighted the long-tailed distribution of relation labels in the primary dataset used in SGG, the Visual Genome dataset (Krishna et al., 2017). In fact, sophisticated neural network techniques barely outperform simple guessing on the most frequent relations (Zellers et al., 2018). Since then, the SGG research community has focused on rectifying this bias. (Guo et al., 2021) proposed a simple yet effective method to re-balance the distribution of relation labels in the training dataset. They also proposed a logit adjustment method to further improve the performance of the SGG model. (Zareian et al., 2020) proposed a dual-scene graph-knowledge graph to incorporate external knowledge into the relational reasoning process. Total Direct Effect (TDE) (Tang et al., 2020) applied causal reasoning to tease out the unbiasing contribution of visual features from potential semantic guessing. We continue on this path of unbiasing SGG by building on the seminal work of BPL-SA (Guo et al., 2021). However, our approach differs greatly from BPL-SA. Instead of a pseudo-domain transfer process, we propose a *bona fide* augmentation for the tail relations in the training dataset.

### 2.2 DATA AUGMENTATION METHODS FOR COMPUTER VISION

Data augmentation has been successfully applied to Computer Vision tasks such as object detection and object localization. In particular, MixUp (Zhang et al., 2018) generates synthetic training images by combining random image pairs from the training data. Specifically, MixUp creates a weighted combination of image features and labels as a new example. CutOut (DeVries & Taylor, 2017) augments and regularizes images by randomly obscuring square regions of the input. Its purpose is to enhance the resilience and overall effectiveness of convolutional neural networks. The primary inspiration behind CutOut stems from the challenge of object occlusion, which is a frequent occurrence in various computer vision applications like object recognition, tracking, and human pose estimation. By generating new images that replicate occluded scenarios, we not only enhance the model's readiness for real-world occlusions but also encourage it to consider a broader context of the image when making decisions. CutMix (Yun et al., 2019) is a technique used to augment image data. Unlike Cutout, which removes pixels, CutMix replaces the removed areas with a patch extracted from a different image. Ground-truth labels are also mixed proportionally based on the number of pixels in the combined images. When these patches are introduced, the localization capability of the model is further improved, since it must now identify objects from partial viewpoints. Our work is inspired by MixUp and CutMix. However, unlike the visually concrete problems of object detection and object localization, our problem uniquely requires abstract relational reasoning about the visual objects. Additionally, we are applying augmentation not as a general regularization technique but as a targeted solution to the long-tailed relational label problem in SGG. To this end, we propose a novel method to augment the tail relations in scene graphs by using visual features and labels.

### 2.3 DATA AUGMENTATION METHODS FOR SCENE GRAPH GENERATION

Although the Balanced Predicate Transfer (BPL) approach (Guo et al., 2021) is related to data augmentation, it does not directly target tail relations. Instead, it rebalances the distribution of relation labels by downsampling triplets with common relation labels. GAN-Neigh (Knyazev et al., 2021) considers a different SGG problem. Instead of relation labels, it tries to balance the biased *triplet* distribution to improve zero-shot generalization performance. Because triplet diversity depend on the both objects and relations, GAN-Neigh augment triplets by perturbing the objects. The GAN component uses a generative adversarial network to create new visual samples in the visual embedding

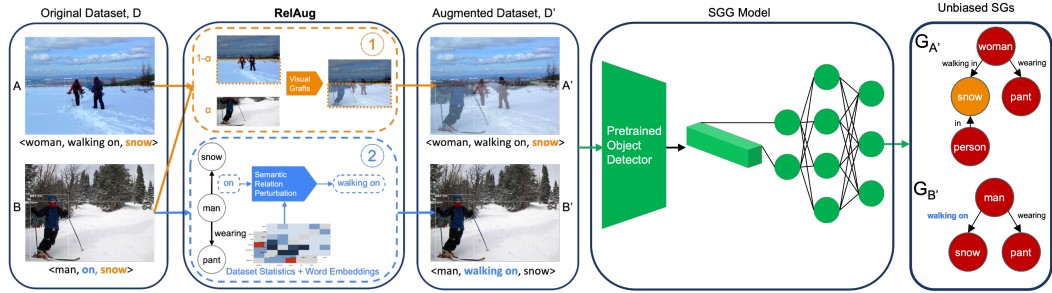

Figure 2: The proposed RelAug pipline corrects the long-tail bias of the original dataset $D$ by augmenting the sample sizes of tail relation labels. RelAug creates new examples using both visual and semantic features. Module (1) describes our Visual Grafts technique, which creates a new image by mixing a source ROI with a target ROI of the same semantic label. Module (2) illustrates a semantic relation perturbation technique, Semantic Grafts, which generates new samples with more informative relation labels. The model learns to generate undersampled relation labels that are semantically related to the oversampled relations. We apply the augmented dataset $D'$ to training any common SGG model to generate unbiased scene graphs.

space. Our motivation and methods differ from those of both BPL and GAN-Neigh. We attack the problem head-on by upsampling the tail relation labels.

## 3 RELATION AUGMENTATION (**RELAUG**) FOR SCENE GRAPH GENERATION

We first introduce formal definitions for the scene graph generation problem in §3.1, before describing the visual (§3.2) and semantic (§3.3) components of our relation augmentation method, **RelAug**.

### 3.1 DEFINITIONS AND OVERVIEW

A scene graph generation (SGG) dataset consists of $N$ tuples, represented as $\mathcal{D} = \{(I, \mathcal{G}, B)\}^N$. Each image $I$ corresponds to its bounding boxes $B$ and its scene graph $\mathcal{G}$. Figure 4 illustrates a scene graph $\mathcal{G} = \{\mathbf{o}, \mathbf{r}\}$ defined by $N = 4$ objects $\mathbf{o} = \{o_1, \ldots, o_{n=4}\}$ and $m = 3$ relationships $\mathbf{r} = \{r_1, \ldots, r_{m=3}\}$ between these objects. Every object $i$ is associated with a bounding box $b_i \in \mathbb{R}^4$ and is categorized as $o_i \in \mathcal{C}$. Each relation can be represented as $r = (o_x, r_k, o_y)$, which denotes a triplet comprising a subject category $o_x$, an object category $o_y$, and a predicate category $r_k \in \mathcal{R}$, where $\mathcal{R}$ is the set of all predicate classes. A typical scene graph generation pipeline first detects objects $\mathbf{o}$ in an image $I$ using a pre-trained object detector such as Faster R-CNN (Ren et al., 2017). Then, as the main objective of SGG, the pipeline determines the relationship predicates $\mathbf{r}$ between each pair of detected object instances. Our objective is to optimize the probability $\Pr(\mathbf{r} \mid \mathbf{o} \times \mathbf{o}; \theta)$, where $\theta$ are the learnable parameters of the SGG model.

A challenge in SGG is the long-tailed distribution of the relation labels in the dataset $\mathcal{D}$ such as Visual Genome (Krishna et al., 2017). Long tails can be modeled by a power-law distribution, $p(x) = Cx^{-\alpha}$, where some relations dominate. As illustrated in Figure 1, common relation labels such as "on" and "with" account for 60% of all label occurrences. To address this imbalance, we employ a grafting-based method to augment $\mathcal{D}$, artificially increasing the occurrence of infrequent relation labels. Our strategy revolves around creating an additional dataset $\hat{\mathcal{D}}' = \{(\hat{I}', \hat{\mathcal{G}}', \hat{B}')\}^{\hat{N}'}$. Recall that $\mathcal{D} = \{(I, \mathcal{G}, B)\}^N$, where the scene graph for each image $I$ is $\mathcal{G} = \{\mathbf{o}, \mathbf{r}\}$. Our model first creates a new set of images $\hat{I}'$ (detailed in §3.2). For each new image, we generate new relations $\mathbf{r}'$ (explained in §3.3). Figure 2 illustrates our entire pipeline. Figure 1 demonstrates the unbiasing effect of our RelAug augmentation.

### 3.2 RELATION AUGMENTATION WITH VISUAL GRAFTS

Pixel-level visual data augmentation techniques, such as MixUp and CutMix (Zhang et al., 2018; Yun et al., 2019), have shown empirical effectiveness for object detection. However, they cannot be

easily applied to the scene graph generation problem, which requires abstract relational reasoning, that is, classifying the relationship between two objects or image regions. Admittedly, visual relations are secondary semantic representations that do not have a clear visual correspondence to the image. After all, what exactly does "standing on" look like? Conveniently, we do not have to solve this relation-to-pixel augmentation directly. Instead, we could proxy a relation-to-pixel augmentation by upsampling triplets containing such relations. Because a relation triplet contains both objects and relations, we can achieve relation augmentation by augmenting the objects in triplets. Intuitively, we can upsample objects through creating new visual instances of the same object categories, retaining the same object and relation labels.

Given the source dataset $\mathcal{D} = \{(I, \mathcal{G}, B)\}^N$, our first step for relation augmentation is to generate examples with (Figure 2 visual features. We first extract source ROIs from the dataset $\mathcal{D} = \{(I, \mathcal{G}, B)\}^N$. Across all images, we build a lookup table $\mathcal{A}$ for bounding boxes with their object labels, where

$$\mathcal{A} = \{(o, b, I) \mid \mathcal{G} = \{\mathbf{o}, \mathbf{r}\} \, \forall \, \mathcal{G}, I, b \in B\} \tag{1}$$

.

Then, for each triplet $\hat{r}$ containing a rare relation $\hat{r_k}$, that is,

$$\hat{r} \in r_k = \{r \mid \text{rank}(r) > k \, \forall \, r \in \mathcal{R}\} \tag{2}$$

, we choose another object that matches the existing subject or object. We make a copy of $I$ with the same $\{B, G\}$, $I'$. However, we replace $I$'s pixels within $b$ with those of another. That is,

For simplicity, a target entry $(\hat{o}, \hat{b}, \hat{I})$ is chosen at random:

$$(\hat{o}, \hat{b}, \hat{I}) \in \{I \mid o' = o \, \forall (o, \mathbf{b}, I) \in \mathcal{A}\} \tag{3}$$

.

The new example image $I'$ can be expressed in terms of its pixel values within the indices $\mathbf{x}$:

$$I'(\mathbf{x}) = \begin{cases} \hat{I}(\mathbf{x}) & \mathbf{x} < \mathbf{b} \\ I(\mathbf{x}) & \text{otherwise} \end{cases} \tag{4}$$

However, because $b \neq \hat{b}$, we need to align the borrowed ROI $I(b)$ with $\hat{I}(\hat{b})$. We express the resizing operation as $\frac{b'}{b}$ and update Equation 4 as

$$I'(\mathbf{x}) = \begin{cases} \dfrac{\mathbf{b}}{\hat{\mathbf{b}}} \hat{I}(\mathbf{x}) & \mathbf{x} < \mathbf{b} \\ I(\mathbf{x}) & \text{otherwise} \end{cases} \tag{5}$$

Because we want a mix up to smooth the pixel values, we want to use a smoothed mixup controlled by a hyperparameter, $\alpha$. Finally, we express a new sample as $(I', \mathcal{G}, B)$ with $\mathcal{G} = (o, r)$, where

$$I'(\mathbf{x}) = \begin{cases} \alpha \dfrac{\mathbf{b}}{\hat{\mathbf{b}}} \hat{I}(\mathbf{x}) + (1 - \alpha) I(\mathbf{x}) & \mathbf{x} < \mathbf{b} \\ I(\mathbf{x}) & \text{otherwise} \end{cases} \tag{6}$$

Visual augmentation selectively chooses $I'$ containing rare relations by repeating the same $\mathcal{G}$.

### 3.3 Relation augmentation with semantic grafts

The visual augmentation generate new samples $(I', \mathcal{G}, B)$ by selectively choosing an $I'$ that contains rare relations repeated in the scene graph $\mathcal{G}$. To further increase the presence of underrepresented relations in the dataset, we modify the sample scene graph $\mathcal{G}$ instead of targeted repetition. Our objective is to construct a new sample with $(I', \mathcal{G}', B)$ with synthetic $\mathcal{G}$ that will produce a more uniform distribution of relation labels. To construct $\mathcal{G}'$, we choose a strategy of perturbing the original $\mathcal{G}$ from $\mathcal{D}$. Recall from §3.1 that $\mathcal{G} = \{(o, r)\}$. Our perturbation schemes take a training scene graph

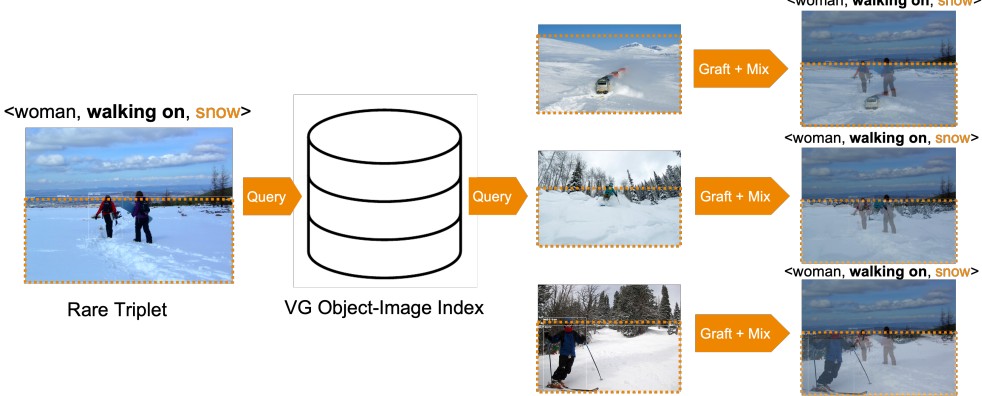

Figure 3: Our relation augmentation strategy repeats a triplet with a rare relation by *visually grafting* another instance of the same object. For example, "walking in" is a rare relation. We replace the original "snow" ROI with another from the dataset to make a new example with the same labels.

$\mathcal{G}$ and perturb it into $\mathcal{G}'$. We focus only on perturbing relations, as it allows the rebalancing of the relation label distribution, so $\mathcal{G}' = \{(o', r')\}$, where $r\hat{} = \{r'_1, \ldots, r'n\}$ are the replacement relation categories. Figure 4 illustrates this scene graph-oriented approach to relation augmentation. However, a simple random perturbation may not generalize to unseen datasets because the generated relation may be implausible. Therefore, we consider several criteria to select only plausible augmentation candidates. We consider the following perturbation methods:

**RAND** (random) is a naive approach as a control. For a given edge $r_i$, we uniformly sample a new relation $r'$ from the top-k rarest relations regardless of the original source relation label $r_i$:

$$r' \in \{r | \operatorname{rank}(r) \geq |\mathcal{R}| - k \ \forall \ r \in \mathcal{R}\} \tag{7}$$

**SIM** (semantically similar) leverages pre-trained GloVe word embeddings $W$ (Pennington et al., 2014) available for each of the relation categories $\mathcal{R}$. Thus, given the edge $r_i$, we retrieve a semantic neighbor of $r_i$ in the embedding space that has a cosine similarity above a threshold $\delta$. We then uniformly sample $r'$ from plausible neighbors of $r_i$ which is also a top-k rarest relation:

$$r' \in \{r | sim(r, r_i, W) \geq \delta, \ \operatorname{rank}(r) \geq |\mathcal{R}| - k \ \forall \ r \in \mathcal{R}\} \tag{8}$$

**CSK** (commonsense knowledge) takes advantage of the commonsense knowledge $K$ given by ConceptNet (Speer et al., 2017). Specifically, we construct another measure of similarity among the relation categories based on their ontological correlations (Zareian et al., 2020). Similar to SIM,

$$r' \in \{r | sim(r, r_i, K) \geq \delta, \ \operatorname{rank}(r) \geq |\mathcal{R}| - k \ \forall \ r \in \mathcal{R}\} \tag{9}$$

**COV** (covariance) uses of the cooccurrence statistics of the Visual Genome (Krishna et al., 2017) training set computed by (Zareian et al., 2020). In contrast to RAND and SIM, COV also depends on the subject node $o_s$ and the object node $o_o$P

$$r' \in \{r | |cov(o_s, r, o_i)| \geq \delta, \ \operatorname{rank}(r) \geq |\mathcal{R}| - k \ \forall \ r \in \mathcal{R}\} \tag{10}$$

The semantic perturbation of the relation labels helps to achieve a reblanced synthetic sample $\mathcal{D}' = (I', \mathcal{G}', B)$ by perturbing the scene graph $\mathcal{G}$ itself. Finally, we combine the modified triplets $\mathcal{D}'$ with the original training set $\mathcal{D}$ to make a combined training set $\mathcal{D} + \mathcal{D}'$. In the example in Figure 4, for an existing triple (visualized as an edge in the scene graph), we augment the original scene graph with an additional rare edge label "walking in" besides the common "in."

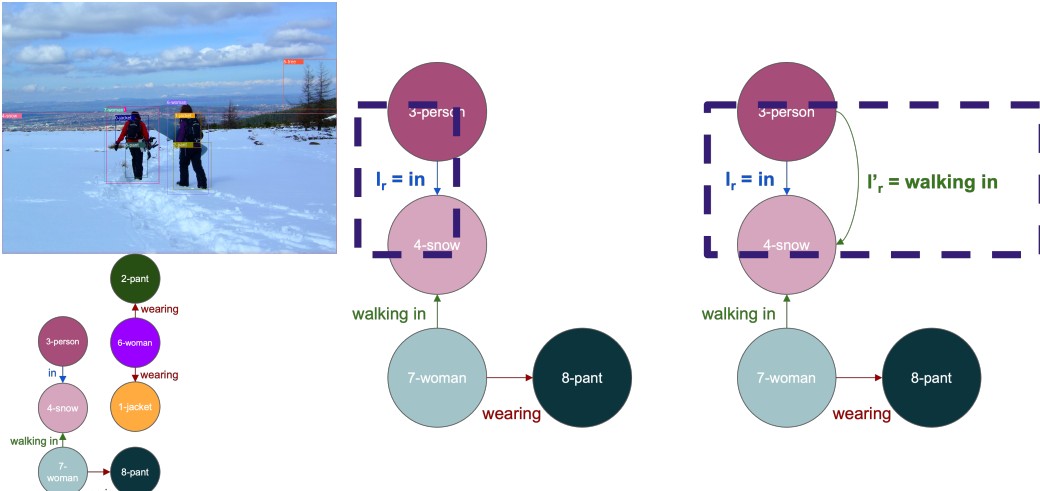

Figure 4: The semantic component of our Relational Augmentation strategy (RelAug) uses semantic relationship between relation labels. Operating on an existing triplet, we add a plausible relation (visualized as an edge in the scene graph) to the original scene graph.

# 4 EXPERIMENTS

## 4.1 DATASETS

The original dataset $D$ is VG150 (Xu et al., 2017), a subset of Visual Genome (Krishna et al., 2017), to evaluate our relation augmentation method. VG150 has 108,077 images. To ensure consistency in experimental results, we follow (Tang et al., 2019)'s data split of the 70%/30% or 57,723/26,446 train/test data split with 5000 validation samples held out from the training split. Consistent with data augmentation practices, we do not augment validation or test datasets.

## 4.2 SETTINGS

**Protocols** We evaluate the performance of the proposed approach on the following three SGG protocols:

- **Predicate Classification (PredCls)**: output only the relation given the bounding boxes and the object labels. This is the easiest setting, and the probability of PredCls corresponds exactly to $P(R|I, B, O)$.

- **Scene Graph Classification (SGCls)** outputs not only the relation but also the objects given only the bounding boxes. The probability of SGCls is given by $P(O|I, B) * P(R|I, B, O)$

- **Scene Graph Generation (SGGen) or Scene Graph Detection (SGDet)**: do all three tasks: bounding box regression, object detection, and relation detection, given no extra ground-truth information. The probability of SGGen is given by $P(B|I) * P(O|I, B) * P(R|I, B, O)$

**Metrics** We mainly consider the mean top K recall for $K \in (20, 50, 100)$. Mean recall, introduced by (Tang et al., 2019; Chen et al., 2019) to the SGG task measures the average per-class recall among all relation categories. Although earlier SGG works such as IMP (Xu et al., 2017) and MotifNet (Zellers et al., 2018) report recall measures, recent works such as BPL-SA (Guo et al., 2021) only report mean recall measures. As Section 1 discusses, given the class imbalance issue in SGG, a model can trivially achieve high recall by guessing the most frequent relations.

**Model zoo** We evaluated three backbone models: Transformer (Tang et al., 2020), MotifNet (Zellers et al., 2018), and VCTree (Tang et al., 2019) and the BPL unbiasing technique (Guo et al., 2021). We follow their original hyperparameter selection.

**Implementation details** We use a learning rate of 0.01 for all three tasks regardless of the number of GPUs for the base model training. We use 0.01 for the Balanced Predicate Transfer (BPL) (Guo et al., 2021) transfer learning.

Table 1: Our RelAug method improves the mean recall performance of classic SGG methods such as Transformer (Tang et al., 2020), MotifNet (Zellers et al., 2018), and VCTree (Tang et al., 2019).

| Model | PredCls | | | SGCls | | | SGGen | | |
|---|---|---|---|---|---|---|---|---|---|
| | mR@20 | mR@50 | mR@100 | mR@20 | mR@50 | mR@100 | mR@20 | mR@50 | mR@100 |
| Transformer (Tang et al., 2020) | 12.4 | 16.0 | 17.5 | 7.7 | 9.6 | 10.2 | 5.3 | 7.3 | 8.8 |
| Transformer+**RelAug** (Ours) | **17.1** | **21.9** | **23.9** | **9.5** | **11.7** | **12.7** | **6.2** | **8.7** | **10.4** |
| Transformer+BPL-SA (Guo et al., 2021) | 26.7 | 31.9 | 34.2 | 15.7 | 18.5 | 19.4 | 11.4 | 14.8 | 17.1 |
| Transformer+**RelAug**+BPL-SA (Ours) | **28.6** | **34.3** | **36.7** | **18.0** | **21.5** | **22.6** | **12.6** | **16.7** | **19.1** |
| MotifNet (Zellers et al., 2018; Guo et al., 2021) | 11.5 | 14.6 | 15.8 | 6.5 | 8.0 | 8.5 | 4.1 | 5.5 | 6.8 |
| MotifNet+**RelAug** (Ours) | **16.5** | **20.8** | **22.8** | **10.2** | **12.6** | **13.3** | **6.5** | **9.2** | **11.1** |
| MotifNet+BPL-SA (Guo et al., 2021) | 24.8 | 29.7 | 31.7 | 14.0 | 16.5 | 17.5 | 10.7 | 13.5 | 15.6 |
| MotifNet+**RelAug**+BPL-SA (Ours) | **26.3** | **32.0** | **34.4** | **15.4** | **18.5** | **19.5** | **11.2** | **15.0** | **17.6** |
| VCTree (Tang et al., 2019) | 11.7 | 14.9 | 16.1 | 6.2 | 7.5 | 7.9 | 4.2 | 5.7 | 6.9 |
| VCTree+**RelAug** (Ours) | **17.7** | **22.2** | **24.2** | **12.1** | **14.8** | **15.9** | **6.4** | **8.8** | **10.5** |
| VCTree+BPL-SA (Guo et al., 2021) | 26.2 | 30.6 | 32.6 | 17.2 | 20.1 | 21.2 | 10.6 | 13.5 | 15.7 |
| VCTree+**RelAug**+BPL-SA (Ours) | **28.7** | **33.9** | **36.6** | **19.6** | **23.3** | **24.7** | **11.0** | **14.5** | **16.8** |

Table 2: Our RelAug method improves the mean recall performance of both classic and recent SGG methods such as DLFE (Chiou et al., 2021) and BPL-SA (Guo et al., 2021). We show the test results in terms of top 20, 50, and 100 mean triplet recalls with graph constraint for PredCls, SGCls, and SGGen. The numbers are in percentage. The highest-performing method for each metric is in bold.

| Model | PredCls | | | SGCls | | | SGGen | | |
|---|---|---|---|---|---|---|---|---|---|
| | mR@20 | mR@50 | mR@100 | mR@20 | mR@50 | mR@100 | mR@20 | mR@50 | mR@100 |
| IMP+ (Xu et al., 2017; Chen et al., 2019) | - | 9.8 | 10.5 | - | 5.8 | 6.0 | - | 7.3 | 8.8 |
| FREQ (Zellers et al., 2018; Tang et al., 2019) | 8.3 | 13.0 | 16.0 | 5.1 | 7.2 | 8.5 | 4.5 | 6.1 | 7.1 |
| KERN (Chen et al., 2019) | - | 17.7 | 19.2 | - | 9.4 | 10.0 | - | 6.4 | 7.3 |
| GPS-Net (Lin et al., 2020) | - | - | 22.8 | - | - | 12.6 | - | - | 9.8 |
| GB-Net (Zareian et al., 2020) | - | 22.1 | 24.0 | - | 12.7 | 13.4 | - | 7.1 | 8.5 |
| VTransE+TDE (Zhang et al., 2017; Tang et al., 2020) | 18.9 | 25.3 | 28.4 | 9.8 | 13.1 | 14.7 | 6.0 | 8.5 | 10.2 |
| Transformer (Tang et al., 2020) | 12.4 | 16.0 | 17.5 | 7.7 | 9.6 | 10.2 | 5.3 | 7.3 | 8.8 |
| Transformer+BPL-SA (Tang et al., 2020; Guo et al., 2021) | 26.7 | 31.9 | 34.2 | 15.7 | 18.5 | 19.4 | 11.4 | 14.8 | 17.1 |
| MotifNet (Zellers et al., 2018; Guo et al., 2021) | 11.5 | 14.6 | 15.8 | 6.5 | 8.0 | 8.5 | 4.1 | 5.5 | 6.8 |
| MotifNet+TDE (Tang et al., 2020) | 18.5 | 24.9 | 28.3 | 11.1 | 13.9 | 15.2 | 6.6 | 8.5 | 9.9 |
| MotifNet+DLFE (Chiou et al., 2021) | 22.1 | 26.9 | 28.8 | 12.8 | 15.2 | 15.9 | 8.6 | 11.7 | 13.8 |
| MotifNet+BPL-SA (Guo et al., 2021) | 24.8 | 29.7 | 31.7 | 14.0 | 16.5 | 17.5 | 10.7 | 13.5 | 15.6 |
| VCTree (Tang et al., 2019) | 11.7 | 14.9 | 16.1 | 6.2 | 7.5 | 7.9 | 4.2 | 5.7 | 6.9 |
| VCTree+TDE (Tang et al., 2020) | 18.4 | 25.4 | 28.7 | 8.9 | 12.2 | 14.0 | 6.9 | 9.3 | 11.1 |
| VCTree+DLFE (Chiou et al., 2021) | 20.8 | 25.3 | 27.1 | 15.8 | 18.9 | 20.0 | 8.6 | 11.8 | 13.8 |
| VCTree+BPL-SA (Guo et al., 2021) | 26.2 | 30.6 | 32.6 | 17.2 | 20.1 | 21.2 | 10.6 | 13.5 | 15.7 |
| Transformer+**RelAug**+BPL-SA (Ours) | **28.6** | **34.3** | **36.7** | **18.0** | **21.5** | **22.6** | **12.6** | **16.7** | **19.1** |
| MotifNet+**RelAug**+BPL-SA (Ours) | **26.3** | **32.0** | **34.4** | **15.4** | **18.5** | **19.5** | **11.2** | **15.0** | **17.6** |
| VCTree+**RelAug**+BPL-SA (Ours) | **28.7** | **33.9** | **36.6** | **19.6** | **23.3** | **24.7** | **11.0** | **14.5** | **16.8** |

## 4.3 DISCUSSION

### 4.3.1 COMPARISONS TO BASELINES AND PRIOR METHODS

As demonstrated in table 1, we show that our RelAug method improves upon all three baseline backbones: Transformer (Tang et al., 2020), MotifNet (Zellers et al., 2018), and VCTree (Tang et al., 2019). For Transformer, we have an average improvement of 4.1 for PredCls, 3.2 for SGCls, and 2.1 for SGGen. We also outperform all previous unbiased SGG methods as shown in Table 1.

### 4.3.2 COMPARISONS TO RECENT UNBIASED SGG METHODS

In addition to our immediate baselines, we achieve superior unbiasing performance as shown in Table 2. Our RelAug method not only outperforms other classic SGG approaches such as IMP+ (Xu et al., 2017), FREQ (Tang et al., 2019), KERN (Chen et al., 2019), . We also outperform recent unbiased SGG approaches icluas GPS-Net (Lin et al., 2020), GB-Net (Zareian et al., 2020), TDE (Tang et al., 2020), and DLFE (Chiou et al., 2021). Most notably, we introduce a nontrivial 3-4 mean recall improvement over our unbiasing technique baseline, BPL-SA (Guo et al., 2021).

### 4.3.3 ABLATION STUDY ON THE VISUAL AND SEMANTIC AUGMENTATION METHODS

Although our final approach includes both semantic and visual triplet augmentation strategies, we investigate their individual contributions to unbiasing scene graph generation (SGG). We apply each component separately and measure each one's mean recall improvement over our baseline neural network architectures: Transformer (Tang et al., 2020), MotifNet (Zellers et al., 2018), and VCTree Tang et al. (2019). As shown in Table 3, not only do both components contribute to a significant increase in mean recall. They bring about a synergistic effect as a whole. Even though they both augment the number of triplets featuring rare relations, they do not conflict with each other and lead to generalization loss. This proves that the generated synthetic examples are sufficiently plausible.

Table 3: We show that both our semantic and our visual augmentation components improve the mean recall performance of our baseline algorithms Transformer (Tang et al., 2020), MotifNet (Zellers et al., 2018), and VCTree Tang et al. (2019). Combined, they produce a synergistic improvement.

| Model | PredCls | | | SGCls | | | SGGen | | |
|---|---|---|---|---|---|---|---|---|---|
| | mR@20 | mR@50 | mR@100 | mR@20 | mR@50 | mR@100 | mR@20 | mR@50 | mR@100 |
| Transformer (Tang et al., 2020) | 12.4 | 16.0 | 17.5 | 7.7 | 9.6 | 10.2 | 5.3 | 7.3 | 8.8 |
| Transformer+**Semantic** (Ours) | 13.0 | 16.7 | 18.2 | 8.0 | 9.9 | 10.5 | 5.6 | 7.8 | 9.4 |
| Transformer+**Visual** (Ours) | 14.7 | 18.5 | 20.2 | 9.0 | 11.0 | 11.8 | 6.2 | 8.4 | 9.9 |
| Transformer+**Semantic+Visual** (Ours) | 17.1 | 21.9 | 23.9 | 9.5 | 11.7 | 12.7 | 6.2 | 8.7 | 10.4 |
| Transformer+BPL-SA (Guo et al., 2021) | 26.7 | 31.9 | 34.2 | 15.7 | 18.5 | 19.4 | 11.4 | 14.8 | 17.1 |
| Transformer+**Semantic**+BPL-SA (Ours) | 27.7 | 33.7 | 35.9 | 17.3 | 20.5 | 21.7 | 12.0 | 15.9 | 18.5 |
| Transformer+**Visual**+BPL-SA (Ours) | 27.7 | 33.3 | 35.5 | 18.0 | 20.7 | 20.7 | 12.7 | 16.4 | 19.1 |
| Transformer+**Semantic+Visual**+BPL-SA (Ours) | 28.6 | 34.3 | 36.7 | 18.0 | 21.5 | 22.6 | 12.6 | 16.7 | 19.1 |
| MotifNet (Zellers et al., 2018; Guo et al., 2021) | 11.5 | 14.6 | 15.8 | 6.5 | 8.0 | 8.5 | 4.1 | 5.5 | 6.8 |
| MotifNet+**Semantic** (Ours) | 15.3 | 19.5 | 21.4 | 7.8 | 9.8 | 10.6 | 5.8 | 8.1 | 10.3 |
| MotifNet+**Visual** (Ours) | 14.5 | 18.4 | 20.1 | 8.6 | 10.8 | 11.6 | 6.0 | 8.5 | 10.1 |
| MotifNet+**Semantic+Visual** (Ours) | 16.5 | 20.8 | 22.8 | 10.2 | 12.6 | 13.3 | 6.5 | 9.2 | 11.1 |
| MotifNet+BPL-SA (Guo et al., 2021) | 24.8 | 29.7 | 31.7 | 14.0 | 16.5 | 17.5 | 10.7 | 13.5 | 15.6 |
| MotifNet+**Semantic**+BPL-SA (Ours) | 25.9 | 31.2 | 33.5 | 14.9 | 17.8 | 18.8 | 10.8 | 14.6 | 16.9 |
| MotifNet+**Visual**+BPL-SA (Ours) | 26.0 | 31.4 | 33.8 | 15.0 | 17.9 | 19.1 | 10.9 | 14.6 | 17.0 |
| MotifNet+**Semantic+Visual**+BPL-SA (Ours) | 26.3 | 32.0 | 34.4 | 15.4 | 18.5 | 19.5 | 11.2 | 15.0 | 17.6 |
| VCTree (Tang et al., 2019) | 11.7 | 14.9 | 16.1 | 6.2 | 7.5 | 7.9 | 4.2 | 5.7 | 6.9 |
| VCTree+**Semantic** (Ours) | 15.8 | 19.7 | 21.5 | 11.4 | 14.0 | 14.9 | 6.2 | 8.7 | 10.3 |
| VCTree+**Visual** (Ours) | 14.6 | 18.6 | 20.3 | 10.1 | 12.7 | 13.5 | 5.5 | 7.8 | 9.4 |
| VCTree+**Semantic+Visual** (Ours) | 17.7 | 22.2 | 24.2 | 12.1 | 14.8 | 15.9 | 6.4 | 8.8 | 10.5 |
| VCTree+BPL-SA (Guo et al., 2021) | 26.2 | 30.6 | 32.6 | 17.2 | 20.1 | 21.2 | 10.6 | 13.5 | 15.7 |
| VCTree+**Semantic**+BPL-SA (Ours) | 27.1 | 31.7 | 33.5 | 18.0 | 30.0 | 22.1 | 10.8 | 13.8 | 16.4 |
| VCTree+**Visual**+BPL-SA (Ours) | 27.3 | 31.8 | 33.9 | 18.2 | 30.5 | 22.7 | 10.9 | 14.0 | 16.4 |
| VCTree+**Semantic+Visual**+BPL-SA (Ours) | 28.7 | 33.9 | 36.6 | 19.6 | 23.3 | 24.7 | 11.0 | 14.5 | 16.8 |

## 5 CONCLUSION

We are the first to propose strategies to augment visual relations. We show that our augmentation strategies address the biased distribution of relations in the dataset and the semantic space. We propose a relation augmentation method to construct a rebalanced relation label distribution. We introduce semantic and visual grafting techniques to create triplets with tail relation labels by increasing both in the image space and in the scene graph space. We demonstrate that this method can be easily adapted to existing methods and produces state-of-the-art performance on the Visual Genome dataset. The impact of our work is the potential to apply this relation augmentation to other visual reasoning tasks, such as Visual Relation Detection (VRD) and Human Object Interaction (HOI). The authors will make the source code publicly available for reproduction.

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
