# OpenReview forum: "A Label is a Label is a Label: Relation Augmentation for Scene Graph Generation"
_ICLR.cc/2024/Conference — ICLR 2024 Conference Withdrawn Submission_

### Official Review · Reviewer_qnbd · 2023-10-23

**Soundness:** 2 fair
**Presentation:** 2 fair
**Contribution:** 2 fair
**Rating:** 5
**Confidence:** 3

**Summary:**

This paper proposes a relation augmentation method called RelAug to address the long-tailed distribution of relation predicates in scene graph generation. The key ideas explored in the paper are: visually grafting to generate new images by mixing object ROIs of the same class and semantic grafting to perturb relation labels in scene graphs using similarity, co-occurence etc. Extensive experiments on backbones such as Transformer, VCTree and MotifNet show consistent gains in performance on mean recall when trained with the combination of the augmented dataset.

**Strengths:**

**Originality**

A novel formulation of directly augmenting relation labels to address long-tail in SGG, unlike prior work which focus on augmenting triplets to address bias in scene graphs.  The technique of semantic and visual grafting has not been explored before to the best of my knowledge.

**Quality and Clarity**

Motivation and proposed approach are clearly described. The relation augmentation pipeline and techniques are easy to follow. Writing is clear and consistent.

**Significance**

Addresses the important problem of long-tail bias in SGG which limits performance on rare classes. Strong empirical gains over competitive baselines demonstrate the significance of this direction.

**Weaknesses:**

**Significance**

The motivation focuses on long-tail distribution of relations, but does not fully justify and compare with a large body of work that outperforms the given method on all the metrics. (some papers discussed below). Also, the paper just uses a single dataset to compare long-tail while there have been other SGG datasets (GQA-LT, VG8k) that have a longer tail and the impact of augmentation could be useful there too.  Providing more empirical analysis for the benefits of relation augmentation could better highlight its significance.

**Methodology**

1. The description of the visual augmentation method is unclear in places - for example, what is the value of alpha and how it is chosen.

2. For the semantic augmentation, additional analysis and justification for the plausibility criteria (similarity, co-occurrence etc) would be helpful. See the questions section for details.

**Evaluation**

1. More qualitative analysis of generated examples from the augmentation techniques would give intuition about their effects. Do they produce plausible, diverse relations?

2. Comparisons to a wider range of prior art in unbiased SGG is missing would better position the advances. Some recent work that uses augmentations and semantic upsampling are missing such as:

a) RelTransformer (https://arxiv.org/pdf/2104.11934.pdf)
b) CaCao visually prompted model for SGG (https://arxiv.org/pdf/2303.13233.pdf)
c) NARE (https://arxiv.org/pdf/2111.13517.pdf)

**Questions:**

1. Are the different semantic perturbations discussed in sec 3.3 applied together or only one of them is considered? There are no ablation studies or experiments regarding this.

2. I do not really get the motivation behind the title and would recommend making it simpler.

3. Most of the methods take into account the cropped subject-object and union regions as features to calculate the scene graph (especially in the two-stage methods), do you think in such a setup the role of visual grafting is not as relevant as claimed to be?


**Minor**

1. Figure 2 is very small and the labels are not visible.
2. The authors mention in Sec 4.1 that the dataset is VG150, but the results are on VG50. Please correct this if that is the case.

---

> ### Comment · Reviewer_qnbd · 2023-11-23
>
> I keep my original rating as the authors did not respond.

---

### Official Review · Reviewer_Q4wR · 2023-10-25

**Soundness:** 3 good
**Presentation:** 1 poor
**Contribution:** 2 fair
**Rating:** 5
**Confidence:** 4

**Summary:**

This paper proposes relation augmentation method for SGG via semantic and visual perturbations. They proposes two data augmentation methods: relation dataset statistics and visual MixUp and grafting techniques, which might improve the results of the existing methods.

**Strengths:**

1. The idea of relation augmentation from semantic and visual aspects is interesting.
2. The motivation of augmenting the tail relation samples is reasonable.
3. This method can be easily adapted to other SGG methods and improve their performance.

**Weaknesses:**

1. The presentation of the paper and the writing need to be improved. There are some mistakes in typos and style.
2. The main contribution of this paper is just a data augmentation, which has been used in other visual tasks. And when replacing the objects in the scene, how to ensure the relationship labels will not be changed.
3. The experiments and results are not convincing. Compared with some recent methods, the results are not SOTA. The authors should give more comprehensive comparisons and analyses with other SGG methods, such as VETO, NICE, and so on.

**Questions:**

See  Weaknesses.

---

> ### Comment · Reviewer_Q4wR · 2023-11-23
>
> I keep my original rating since the author did not give the response

---

### Official Review · Reviewer_SXQd · 2023-10-29

**Soundness:** 2 fair
**Presentation:** 2 fair
**Contribution:** 2 fair
**Rating:** 3
**Confidence:** 5

**Summary:**

This paper proposes a novel method of generating unbiased scene graphs by augmenting training samples semantically and visually. The semantic augmentation is performed via random selection, semantic similarity, commonsense knowledge, and co-occurrence statistics. The visual augmentation is performed via visual grafting and mixing of rare relationships in query images. The diversity metric mR@K is chosen as the evaluation metric and results demonstrate that the proposed method improves the tail relationships.

**Strengths:**

This paper has the following strengths

**1. Well-motivated:** The motivation for choosing augmentation as a debiasing strategy is clearly explained.

**2. Easy-to-follow.** The papers' writing flow is simple and easy to follow. The methodology of visual and semantic augmentation is clearly written with proper mathematical notation and symbols.

**3. Good Presentation.** Figure 3 and Figure 4 properly describe the semantic and visual augmentation process with good examples.

**4. Improvement of mR@K.** The paper improves the mean recall@K of several classical and recent SGG methods. Their method consistently improves the mR@K in all evaluation settings.

**5 Good ablation study on semantic and visual augmentation.** In Table 3, the authors demonstrate the contribution of semantic and visual augmentation technique for several baselines. It is interesting to see how both semantic and visual augmentation contributes to the overall improvement.

**Weaknesses:**

This paper has the following weaknesses

**1. Evaluation of Recall@K missing.** The augmentation of a rare triplet may hurt the semantically similar common triplet. For example, if we augment the training with 'man-walking-on-snow', it may hurt the performance of 'man-on-snow'. Therefore, it is vital to observe how the recall@K is affected after the augmentation.

**2. Evaluation of zsR@K missing.** A great advantage of augmenting training samples is the improvement of the zero-shot capability of a model. For example, if we create a new training triplet 'boy-riding-snowboard' from the visual and/or semantic augmentation of the original triplet 'boy-on-snowboard, it may coincide with some zero-shot examples in the testing dataset. In general, it should improve the zsR@K metric and it would strengthen the claim of the paper. If the papers' augmentation does not improve the zsR@K, that also requires some investigation and should be reported/discussed.

**3. Per-class bar plot of recall improvement missing:** In Figure 1, authors demonstrated the per-class distribution of relationships before and after augmentation. It would be very interesting if we had a similar bar plot for recall-improvement of each class. Such bar plots are now present in almost every published SGG paper and they provide useful insight into the methods' robustness to different classes. I suspect the performance increase/ decrease of the classes would be proportional to its increase/decrease after the augmentation process. Such a bar plot would also provide the authors with insight on how to efficiently augment the training classes.

**4. Missing ablation study of different semantic augmentation methods.** In section 3.3, different perturbation method of semantic augmentation has been discussed. However, how each of these influences the performance is not demonstrated in the evaluation. Without such ablation studies, readers would find the detailed discussion without any ties to the results section.

**5. Missing ablation study on grafting hyperparameter.** Eqn. (6) describes the smoothed mixup-like augmentation of the visual domain. However, how sensitive the performance is with respect to the hyperparameter $\alpha$ is not studied. The performance should be equal to the baseline model with $\alpha=0$ and maximize to a certain value of $\alpha$. Both recall and mean recall can be observed with this sweep of $\alpha$.

**Questions:**

See weaknesses

**Post-rebuttal rating:** None of my concerns have been addressed during the discussion period and therefore, I retain my original rating of 3.

---

> ### Comment · Reviewer_SXQd · 2023-11-23
> **Final Rating**
>
> None of my concerns have been addressed during the discussion period and therefore, I retain my original rating of 3.

---

### Official Review · Reviewer_vthg · 2023-10-30

**Soundness:** 2 fair
**Presentation:** 2 fair
**Contribution:** 2 fair
**Rating:** 3
**Confidence:** 4

**Summary:**

This paper proposes a relation augmentation strategy (RelAug) for solving the biased problem of scene graph generation. Specifically, RelAug includes two parts: visual augmentation by fusing pixels of objects with the same categories and semantic augmentation by replacing relationship categories with similar semantic meanings. The idea of the paper is intuitive, but some descriptions, including the title and the introduction of the motivation, are a little exaggerated, and the ablation experiments are not sufficient. The technical novelty seems poor.

**Strengths:**

The problem of solving unbiased SGG is meaning and the idea of the paper is reasonable and intuitive.

**Weaknesses:**

The main problem is:
1、 Even though the authors claim that they are the first to propose strategies to augment visual relations, several works have focused on relationship augmentation, such as:

Fast Contextual Scene Graph Generation with Unbiased Context Augmentation

Compositional Feature Augmentation for Unbiased Scene
Graph Generation

Visual compositional learning for human-object interaction detection (HOI task, which is similar to SGG)


2、The equations and images are not well-defined. For example, in Equations (4) and (5), the < should be $\in$. What is x? What is b? It seems like the author uses the same symbol to reflect different meanings; for instance, the b may refer to a box region as well as a box size. The left subfigure of Figure 1 is enough.


3、The experiments are not sufficient. It would be better to give a recall, or there may exist the possibility to increase the mean recall by significantly sacrificing the accuracy of the recall. More state-of-the-art methods need to be compared, for example, unbiased methods in 2022 and 2023. More ablation experiments are required. How to choose $\alpha$ in the equation? How do you select top-k? How to choose the manner of perturbation methods? More analysis should be provided, for example, the exact relationships the proposed strategy improved and more explanation about why RelAug works.

**Questions:**

Please refer to the weakness.

1、The novelty. What are the differences between the proposed work and previous methods? What is the novelty of this paper?

2、 Please revise the definitions of equations and figures.

3、The experiments are not sufficient. More experiments and analysis are needed.

---

### Official Review · Reviewer_o27p · 2023-10-31

**Soundness:** 2 fair
**Presentation:** 3 good
**Contribution:** 2 fair
**Rating:** 3
**Confidence:** 5

**Summary:**

This paper proposes visual and semantic augmentation strategies to help address the bias distribution of scene graph generation. It presents experiments on the visual genome to show its performance.

**Strengths:**

The comparisons with baseline methods are also quite obvious.
Introducing data augmentation to address bias distribution is intuitive.

**Weaknesses:**

I am quite concerned with the novelty. For semantic augmentation, it uses several strategies to perturb the relationships. However, I do not know why these strategies can lead to better performance. It seems it may lead to a mass of noisy labels. I wonder if I missed some points. For visual augmentation, it is similar to MixUp and CutMix except that it mixes the objects of the same category.

The authors merely compare with some methods that were published before 2021. However, more recent works that obtain better performance are intensively proposed. I think these works [1,2] should be included as the baseline to better demonstrate the effectiveness of the proposed algorithm. Besides, The authors merely compare with some baseline, and there exist methods [3] that also address the unbias SGG task. However, these works are also not included for comparisons.

Some minor issues with grammar errors and punctuation.

[1] Prototype-based Embedding Network for Scene Graph Generation, in CVPR, 2022
[2] RU-Net: Regularized Unrolling Network for Scene Graph Generation, in CVPR, 2022
[3] Stacked Hybrid-Attention and Group Collaborative Learning for Unbiased Scene Graph Generation, in CVPR, 2022

**Questions:**

See weakness

---

> ### Comment · Reviewer_o27p · 2023-11-23
>
> It seems no responds are given, and I keep the original rating.